# Antimicrobial Activities of Secondary Metabolites from Model Mosses

**DOI:** 10.3390/antibiotics11081004

**Published:** 2022-07-26

**Authors:** Lia R. Valeeva, Ashley L. Dague, Mitchell H. Hall, Anastasia E. Tikhonova, Margarita R. Sharipova, Monica A. Valentovic, Lydia M. Bogomolnaya, Eugene V. Shakirov

**Affiliations:** 1Institute of Fundamental Medicine and Biology, Kazan Federal University, Kazan 420008, Russia; lia2107@yandex.ru (L.R.V.); aetikhonova@mail.ru (A.E.T.); marsharipova@gmail.com (M.R.S.); 2Department of Biological Sciences, College of Science, Marshall University, Huntington, WV 25701, USA; dague3@marshall.edu (A.L.D.); hall755@marshall.edu (M.H.H.); 3Department of Biomedical Sciences, Joan C. Edwards School of Medicine, Marshall University, Huntington, WV 25755, USA; valentov@marshall.edu

**Keywords:** plant metabolite, Bryophytes, moss, *Physcomitrium patens*, *Sphagnum fallax*, antibacterial activity, exudate, extract

## Abstract

Plants synthetize a large spectrum of secondary metabolites with substantial structural and functional diversity, making them a rich reservoir of new biologically active compounds. Among different plant lineages, the evolutionarily ancient branch of non-vascular plants (Bryophytes) is of particular interest as these organisms produce many unique biologically active compounds with highly promising antibacterial properties. Here, we characterized antibacterial activity of metabolites produced by different ecotypes (strains) of the model mosses *Physcomitrium patens* and *Sphagnum fallax*. Ethanol and hexane moss extracts harbor moderate but unstable antibacterial activity, representing polar and non-polar intracellular moss metabolites, respectively. In contrast, high antibacterial activity that was relatively stable was detected in soluble exudate fractions of *P. patens* moss. Antibacterial activity levels in *P. patens* exudates significantly increased over four weeks of moss cultivation in liquid culture. Interestingly, secreted moss metabolites are only active against a number of Gram-positive, but not Gram-negative, bacteria. Size fractionation, thermostability and sensitivity to proteinase K assays indicated that the secreted bioactive compounds are relatively small (less than <10 kDa). Further analysis and molecular identification of antibacterial exudate components, combined with bioinformatic analysis of model moss genomes, will be instrumental in the identification of specific genes involved in the bioactive metabolite biosynthesis.

## 1. Introduction

The emergence of bacterial drug resistance, especially in hospital settings, represents the next great frontier in healthcare. If no action is taken, diseases caused by antibiotic-resistant bacteria are predicted to kill up to 10 million people a year by 2050, similar to reported mortality rates for cancer [1]. Thus, in 2019, the World Health Organization listed antimicrobial resistance among the ten biggest health challenges [2]. Among different pathogens, Gram-positive bacteria represent particularly serious health concerns, as many key infections are caused by multidrug-resistant Gram-positive bacteria, including methicillin-resistant *Staphylococcus aureus* (MRSA), vancomycin-resistant *Enterococcus faecium* (VRE) and erythromycin-resistant group A Streptococcus (GAS) [3]. The rise of antibiotic-resistant microbial infections brings about further fears that the last remaining drugs to treat Gram-positive bacterial infections may become ineffective. Thus, the critical need to discover new potent antibiotics is becoming widely recognized.

Despite the clear demand for more antimicrobial agents, very few new antibiotics are reaching the market—the last entirely original class of antibiotic was discovered in the late 1980s. One of the critical barriers to progress in the field is that the search for natural antibiotics has been historically limited mostly to soil microorganisms (fungi, bacteria) that can be propagated in the laboratory setting, whereas other phyla remained relatively untapped. At the same time, plants produce a variety of bioactive secondary metabolites, peptides and various small molecules with unique biological functions, including defense from environmental threats and resistance to microbial pathogens [4,5]. Some well-known examples of herbal-based pharmaceuticals include acetylsalicylic acid (isolated from willow bark), artemisinin (an antimalaria drug from Artemisia annua) and Taxol (an anticancer drug from Pacific yew conifer).

There are at least five classes of known secondary metabolites produced by plants: terpenes, aromatics, glucosinolates, benzoxazinoids and green leaf volatiles [6]. In the model flowering plant *Arabidopsis thaliana,* various glucosinolates and benzoxazinoids play important roles in the defense against *Pseudomonas syringae*. Glycoalkaloids, glucosinolates and cannabinoids produced by *Solanum nigrum*, *Armoracia rusticana* and *Cannabis sativa* display antimicrobial activity against Gram-positive and Gram-negative bacteria [6]. Additionally, secreted peptides and small proteins are also involved in plant defense and adaptation to environmental stress [7,8,9]. Therefore, identification and characterization of new plant compounds with antibacterial activity can be a viable route to new antibiotic discovery.

Although some data exist on the presence of bioactive secondary metabolites from the flowering plants (Angiosperms), little is known about secondary metabolites from other groups of the plant kingdom, especially from the early diverging non-vascular plants, currently represented by the ancient Bryophyte division. With over 20 thousand extant species, Bryophytes (mosses, liverworts and hornworts) are the second most diverse group of land plants after flowering plants. Among all terrestrial plants, Bryophytes are considered by many scientists as a nearly unexplored natural reservoir of new biologically active secondary metabolites [10]. Well adapted to different environmental and stress conditions, Bryophytes evolved a number of defense mechanisms, including production of various antimicrobial compounds [11]. In a study of several mosses, high antimicrobial activity was found against fungi and both Gram-negative and Gram-positive bacteria [12]. Metabolites from several other mosses were also shown to effectively inhibit growth of Gram-negative or Gram-positive bacteria [10,13]. In addition to mosses, liverworts, such as *Marchantia polymorpha* and *Conocephalum conicum,* were also described as potent producers of bioactive compounds with antimicrobial activity [14].

Overall, Bryophytes produce a number of unique natural compounds with antimicrobial properties; however, very few of them have been biochemically isolated or characterized in detail [15]. The current inadequacy in their comprehensive analysis stems largely from the small physical size of Bryophytes, significant gaps in their classification [16] and numerous technical obstacles, such as the lack of powerful genomic and proteomic tools for their analysis [17,18]. However, these limitations can be circumvented by the use of model Bryophytes, such as *Physcomitrium*
*patens* (formerly *Physcomitrella patens*) and *Sphagnum fallax,* whose genomes have been completely sequenced [19,20] and standard laboratory techniques for their maintenance, transformation and even growth in bioreactor conditions have been well established [21,22]. Though such model mosses are often used to investigate unique biological characteristics of Bryophytes, such as adaptations to life on land and to drought stress [23], they have previously not been exhaustively studied for antimicrobial potential.

Here, to extend the arsenal of available antimicrobials, we aim to characterize antibacterial activity of intracellular and extracellular metabolites produced by the model mosses *P.*
*patens* and *S. fallax*. We show that both mosses synthetize polar and non-polar intracellular compounds with antimicrobial activity against Gram-negative *Pseudomonas syringae* bacteria. Interestingly, in contrast to intracellular metabolites, secreted water-soluble *P. patens* exudates display specific inhibitory activity against *S. aureus* and other Gram-positive bacteria, but not against Gram-negative bacterial species. Analysis of exudate stability under various physical conditions indicated that secreted metabolites are stable after multiple freezing/thawing cycles and in different light conditions, but their antimicrobial activity is substantially reduced following sample boiling or treatment with the proteinase K enzyme. Furthermore, size fractionation experiments indicated that the bioactive moss compounds are <10 kDa in size. Taken together, our data suggest that secreted antibacterial moss compounds are likely peptides or small proteins. Overall, our approach of using model mosses with completely sequenced genomes for the identification of novel antibacterial compounds will allow future bioinformatic analyses to identify specific genes involved in their biosynthetic pathways.

## 2. Results

### 2.1. Determination of Antibacterial Activity of Intracellular Moss Metabolites

We first tested model moss extracts for putative antibacterial activity displayed by intracellular metabolites. Metabolites were extracted from 10-day-old protonema tissues of the previously sequenced *Physcomitrium patens* ecotype Gransden (Gd) [19] and 30-day-old gametophores of *Sphagnum fallax* strain MW (Table 1). To determine optimal extraction conditions, we employed different combinations of solvents and extraction time: 80% methanol (for polar compounds) and hexane (for non-polar metabolites) extractions for 24 h and 45 h, followed by the qualitative assessment of extracted metabolites for antibacterial activity using the disk-diffusion (DDM) assay (clsi.org).

The 24 h methanol extracts from both *P. patens* and *S. fallax* showed statistically significant (*t*-test, *p* ≤ 0.05) inhibitory activity against the phytopathogenic Gram-negative *P. syringae* DC3000 strain, though activity levels dropped below significance levels for 45 h methanol extracts from both mosses (Figure 1 left panel, Table 2). Hexane extracts from both mosses also displayed antibacterial activity (Figure 1 right panel, Table 2). Interestingly, extending the metabolite extraction time with hexane had a positive effect on antibacterial activity: the strongest inhibition of *P. syringae* growth was observed with 45 h hexane extracts from both *P. patens* Gd and *S. fallax* MW (*t*-test, *p* ≤ 0.01), whereas activity levels of 24 h hexane extracts were not significant (Figure 1 right panel and Table 2). This observation suggests that the ratio of bioactive polar and non-polar compounds changes with extraction time for both mosses.

We also tested a smaller subset of moss extracts against other Gram-negative and Gram-positive bacteria (Table 2). Neither extract showed antibacterial activity against Gram-negative bacterium *Serratia marcescens*, and *P. patens* extracts showed no activity towards Gram-negative *Escherichia coli* or Gram-positive *Staphylococcus aureus* bacteria. Overall, we conclude that the model moss extracts tested in qualitative DDM assays displayed statistically significant antibacterial activity levels against Gram-negative phytopathogenic *P. syringae* bacteria, though this activity appeared relatively unstable in follow-up quantitative assays and was not pursued further.

### 2.2. Identification of Antibacterial Activity in Moss Exudates

Several types of small extracellular peptides and metabolites from mosses and other plants are known to have antibacterial properties [5,9,24]. Thus, two *P. patens* ecotypes, Gransden (Gd) and Villersexel (Vx), were grown in liquid cultures for 1, 2 and 4 weeks and their exudates (water-soluble fractions secreted to the growth medium) were tested for the presence of antimicrobial activity. DDM assays detected substantial antimicrobial activity of *P. patens* exudates against Gram-positive *S. aureus* bacteria (Figure 2A), but not against Gram-negative *Salmonella* Typhimurium or *S. marcescens* (Figure 2B,C). Antibacterial activity of *P. patens* exudates against *S. aureus* was detectable even at the first time point of analysis (1 week of moss growth) and further increased with longer moss growth times (Table 3). Interestingly, antibacterial activity against *S. aureus* in exudates from the Gd ecotype appeared to reach its maximum at 2 weeks of growth (Figure 2D), whereas in exudates from the Vx ecotype this activity continued to increase over time and peaked at 4 weeks of moss growth (Figure 2E). Collectively, our data indicate that extracellular exudates of two different ecotypes of the model moss *P. patens* display high growth inhibitory activity against Gram-positive bacterium *S. aureus* ATCC25293.

### 2.3. Quantitative Analysis of Antibacterial Activities of P. patens Exudate

Although the DDM assay is a very powerful qualitative method for initial detection of antibacterial activity, it is not suitable for quantitative characterization of moss exudates because of the differences in the diffusion capacity of various exudate components in the solid medium. Thus, following CLSI guidelines, we employed a broth microdilution method to determine minimum inhibitory concentration (MIC) of metabolites present in *P. patens* exudates. Since the genome of the *P. patens* Gd ecotype was previously fully sequenced [19], we specifically focused on exudates from the two-week and four-week-old Gd ecotype in MIC assays. First, we performed a series of positive and negative control experiments. As expected for a negative control test, the addition of resuspended BCD medium concentrate alone did not inhibit *S. aureus* culture growth in the Mueller–Hinton medium (Figure 3A). In contrast, the addition of a range of carbenicillin and chloramphenicol antibiotic dilutions inhibited the growth of *S. aureus* (Figure 3B) and other Gram-positive bacteria (Figure 3C,D).

Next, we used lyophilized moss exudates (containing all secreted *P. patens* compounds and inorganic salts from the BCD medium) that were serially diluted to 50, 25, 12.5 and 6.25 mg/mL concentrations in 96-well microtiter plates containing *S. aureus* cultures to evaluate bacterial growth dynamics for 18 h (Figure 4). Interestingly, exudates from two-week-old *P. patens* Gd cultures completely inhibited the growth of *S. aureus* at a 25 mg/mL concentration (Figure 4A), whereas four-week-old *P. patens* Gd exudates displayed a 2-fold lower MIC value of 12.5 mg/mL (Figure 4B). These quantitative MIC data indicate that, unlike the prediction from qualitative DDM assays, antibacterial activity in *P. patens* Gd exudates actually continues to increase over time, supporting the notion that both Gd and Vx ecotypes show apparently similar patterns of antibacterial activity accumulation as their cultures get older.

We next asked if *P. patens* exudates are also active against other Gram-positive bacteria besides *S. aureus*. Specifically, we tested exudates against *Streptococcus pyogenes* and *Enterococcus faecium* strains, which represent close relatives of GAS and VRE bacteria from the “Biggest Threats” CDC list [3]. Indeed, exudates from the four-week-old *P. patens* Gd strain were able to inhibit growth of both pathogens (Figure 5). Interestingly, MIC values were 6.25 mg/mL for *S. pyogenes* (Figure 5A) and 25 mg/mL for *E. faecium* (Figure 5B), indicating that *S. pyogenes* bacteria are more sensitive to the effects of the Gd moss exudate than other tested bacteria and implying that sensitivity to the *P. patens* exudate varies among different Gram-positive bacteria. Finally, we used the MIC assays to confirm DDM data that *P. patens* exudates are not effective against Gram-negative bacteria. Indeed, four-week-old *P. patens* Gd exudates were not effective against *S.* Typhimurium (Figure 5C) or *S. marcescens* (Figure 5D). Taken together, our data indicate that *P. patens* moss secretes potent antimicrobial metabolites with high specificity against Gram-positive bacteria.

### 2.4. Stability of Antibacterial Compounds in Moss Exudates

Analysis of exudate stability under different physical conditions can aid in determining the likely chemical nature of the antibacterial exudate components. Thus, we subjected *P. patens* Gd exudates to different treatment regimens, such as heating, light exposure, proteinase K treatment, repeated freezing and thawing, and evaluated residual antibacterial activity using the MIC assays. As expected, neither treatment affected *P. patens* exudates diluted to a 6.25 mg/mL concentration, which showed no inhibition of *S. aureus* growth (Figure 6A). When used at the MIC concentration of 12.5 mg/mL, *P. patens* exudates fully retained antibacterial activity against *S. aureus* after 3 h exposure to room temperature (RT) with or without direct sunlight, and after repeated freezing and thawing cycles (Figure 6B), indicating that these treatments do not affect antimicrobial activity of the Gd exudate. In a sharp contrast, antibacterial activity was substantially reduced when *P. patens* exudates at MIC values (12.5 mg/mL) were either boiled or treated with proteinase K (Figure 6B). Furthermore, boiling substantially reduced antibacterial activity even at a higher exudate concentration of 25 mg/mL (Figure 6C). Though not fully sufficient to rule out other possibilities, these results are consistent with the peptide or protein nature of secreted antimicrobial *P. patens* metabolites.

### 2.5. Size Fractionation of Bioactive Exudate Metabolites

To determine the approximate molecular weight range of secreted antimicrobial *P. patens* compounds, we performed a size fractionation experiment using the Amicon Ultra-15 ultrafiltration system (3 kDa and 10 kDa molecular weight cutoffs). Following size fractionation, MIC data indicated that antibacterial activity against *S. aureus* was completely lost in the largest >10 kDa fraction (Figure 7A). In contrast, antibacterial activity was fully retained in the smaller 3–10 kDa size fraction (Figure 7B) and also to some degree in the smallest <3 kDa fraction (Figure 7C). Specifically, the MIC value of the most active 3–10 kDa fraction (25 mg/mL) was only 2-fold less than that of the unfractionated Gd exudate (Figure 4B). Overall, these data indicate that the apparent molecular weight of the bioactive *P. patens* exudate compounds is less than 10 kDa, which correlates well with their presumed peptide or small protein nature determined through proteinase K and boiling sensitivity assays.

## 3. Discussion

Bryophytes produce a number of different compounds with unique biological activities [25], though less is known about antimicrobial metabolites from model mosses. Model Bryophytes provide a promising avenue for natural product discovery as they confer several important advantages over other plants, including completely sequenced genomes, well-developed laboratory techniques and transgenic manipulation methods for follow-up biotechnological and metabolomic applications [26,27,28]. Despite some recent advances in model moss proteomics, metabolomes and secretomes of model mosses are not yet extensively studied. Here, we evaluated intracellular and extracellular fractions from *P. patens* and *S. fallax* for the presence of potent antibacterial activity.

To initiate our analysis, we first evaluated intracellular compounds of *P. patens* and *S. fallax* for their antibacterial potential. Both polar (methanol-based) and non-polar (hexane-based) extracts inhibited the growth of Gram-negative *P. syringae* DC3000 bacteria. These findings are in the agreement with results from another study which evaluated extracts from 42 different Bryophytes and detected moderate antimicrobial activity against several bacterial species [29]. Overall, though the presence of antibacterial activity in the methanol and hexane extracts of *P. patens* and *S. fallax* mosses appeared promising, the bioactive metabolites were not stable during subsequent extract processing steps, and completely lost their activities after lyophilization (data not shown). Thus, we focused on the antibacterial potential of secreted exudate fractions, which remained stable throughout the experiments.

Unlike *S. fallax*, all *P. patens* strains can be easily propagated in liquid cultures where they secrete exudates containing a number of extracellular compounds [8,21]. First, we tested exudates from two laboratory *P. patens* strains, Gd and Vx, for their ability to inhibit bacterial growth in disk diffusion assays. Exudates from both *P. patens* strains had high antimicrobial activity against Gram-positive *S. aureus* bacteria, suggesting that the production of antibacterial metabolites is not specific to any one individual strain, but is likely a common feature of different *P. patens* isolates. To get a broader understanding of bioactive moss metabolite specificity, we assayed a panel of different Gram-positive and Gram-negative bacterial species. In addition to *S. aureus* bacteria, *P. patens* exudates showed growth inhibitory activity against two other Gram-positive bacteria, *E. faecium* and *S. pyogenes*. In a stark contrast, no antibacterial activity was observed with any exudates against Gram-negative bacteria *Salmonella*, *S. marcescens* or *E.coli*. Taken together, these data indicate that *P. patens* produces secreted metabolites with antibacterial activity specific only against Gram-positive but not Gram-negative bacteria.

The narrow specificity of moss metabolites only against Gram-positive bacteria is intriguing and may stem from the structural differences in bacterial cell walls or antibiotic resistance mechanisms. For instance, some features of Gram-negative bacteria may make it hard for moss antimicrobials to reach their targets: (a) the presence of the outer and inner membranes and the periplasmic space, which may represent a physical barrier to peptide antibiotics [30]; (b) very efficient proteases and powerful efflux systems which quickly remove antibiotics [31]; and (c) the presence of antimicrobial stress-responding proteins in periplasmic space, such as SipA in *Vibrio cholerae* and SapA in *Actinobacillus pleuropneumoniae* [32,33]. Other, currently unknown, mechanisms may also be involved.

Interestingly, time course experiments indicated that antibacterial activity in *P. patens* exudates increases over time: it is already detectable after 1 week of moss culture growth and continues to increase after 2 weeks and further after 4 weeks. Two possible scenarios can be envisioned that explain accumulation of antibacterial activity in moss exudates over time. First, the bioactive moss compounds may be relatively stable in the environment under the chosen *P. patens* growth conditions and accumulate in the exudate as the moss culture continues to grow. Alternatively, antimicrobial compounds may be relatively unstable and degrade quickly, but their higher abundance in the four-week-old exudate may simply reflect a moss biomass increase over the longer growth period.

To start addressing the question of bioactive moss metabolite stability and to obtain a glimpse into the potential molecular nature of antimicrobial moss compounds, we performed a series of thermo- and photostability assays. Our MIC experiments indicated that *P. patens* exudates fully retained antibacterial activity against *S. aureus* after various physical challenges, but activity was greatly diminished when *P. patens* exudates were either boiled or treated with proteinase K. These results are consistent with the peptide or small protein nature of antimicrobial *P. patens* metabolites. Furthermore, our size fractionation data also indicated that the antibacterial activity is associated with the smaller <10 kDa molecular weight fractions. Thus, the apparently small molecular weight of bioactive *P. patens* exudate compounds correlates well with their presumed peptide nature.

Several types of peptide antibiotics have been described in the literature. One type of such antibiotic represents mostly bacterial polypeptides not synthesized on ribosomes, including polymyxins, bacitracins and glycopeptides [34]. Another type of antimicrobial peptide is ribosomally synthesized by many diverse organisms as an inherent component of the natural host defense system. Though these compounds are often less well characterized, they represent a potentially very promising opportunity for natural antibiotic development. Specifically, *P. patens* is known to produce over 400 secreted proteins, most of which are pathogen defense-related and 72 are secreted only in the presence of chitosan, an elicitor of the plant pathogen response pathway [24]. In addition to known secreted proteins, *P. patens* also exudes over 500 cryptic peptides with presumed antibacterial and regulatory activities [8,9]. However, previous data indicated that the synthesis and secretion of such cryptic peptides are typically induced by phytopathogens, whereas our results indicate that *P. patens* cells constitutively secrete antibacterial compounds when grown under standard laboratory conditions. Thus, identification of such constitutively synthesized metabolites offers an important advantage over pathogen-induced antibacterial peptides, as moss cells in our experiments are not artificially manipulated, have no reduction in growth rate and experience no other unwanted physiological changes during growth in culture.

## 4. Materials and Methods

### 4.1. Moss Strains and Growth Conditions

*S. fallax* strain MW (a gift from Dr. David Weston and Dr. Megan Patel, Oak Ridge National Laboratory) and *P. patens* ecotypes Gransden (Gd) [19] and Villersexel (Vx) (gifts from Dr. Pierre-François Perroud and Dr. Stefan Rensing, Philipps-Universität Marburg) were propagated on Petri plates with BCD agar medium containing 1 mM MgSO_4_, 1.84 mM KH_2_PO_4_ pH 6.5, 10 mM KNO_3_, 0.045 mM FeSO_4_, 1 mM CaCl_2_, and the trace elements of 9.93 mM H_3_BO_3_, 2.2 mM CuSO_4_ × 5H_2_O, 1.96 mM MnCl_2_ × 4H_2_O, 0.231 mM CoCl_2_ × 6H_2_O, 0.191 mM ZnSO_4_ × 7H_2_O, 0.169 mM KI and 0.103 mM Na_2_MoO_4_ × 2H_2_O, supplemented with 5.5 mM ammonium tartrate and 0.7% agar [35]. Moss tissue was passaged weekly by homogenizing with the IKA Ultra-Turrax T10 basic tissue dispenser, followed by plating on cellophane disks placed on solid BCD medium in Petri dishes. Moss plates were grown in a plant growth chamber (Model 7300, Caron Products) at 22 °C, 65% humidity, 880 lux light intensity and 12/12 h light/dark conditions.

### 4.2. Intracellular Metabolite Extraction

Polar and non-polar moss metabolites were extracted with 80% methanol and hexane treatments, respectively. Mosses were grown for 10 days (*P. patens*) or 30 days (*S. fallax*) on Petri dishes with cellophane disks placed on solid BCD agar medium. Moss tissue (1.5 g) was collected with a spatula, excess moisture was removed by blotting with a paper towel and tissue was ground to a thin powder with mortar and pestle (model 29-151, Genesee Scientific) using liquid nitrogen. Tissue powder was transferred into a tube with 15 mL of the appropriate solvent, wrapped in aluminum foil and metabolites were extracted by maceration at RT for 24 or 45 h. Extracts were collected by centrifugation at 4300× *g* for 15 min (centrifuge model 5424R, Eppendorf) and filtered through a 0.45 µm syringe filter. Samples were first concentrated by drying under a stream of nitrogen, and subsequently, fully dried in a lyophilizer (model 7382021, Labconco). Dry pellets were weighted and stored at −80 °C before use. For analysis, samples were dissolved in 80% methanol or hexane at a final concentration of 100 µg/µL.

### 4.3. Preparation f Extracellular Metabolites from Moss Exudates

For the analysis of secreted metabolites, *P. patens* ecotypes were grown in 250 mL flasks containing 100 mL of liquid BCD medium on the orbital shaker (with rotation 150 rpm) at 22 °C, 65% humidity, 880 lux light intensity and 12/12 h light/dark conditions. Moss cultures were grown for 1–4 weeks depending on the experiment, and exudates were collected by filtering first through a 70 µm cell strainer and then through a 0.45 µm syringe filter, and were flash frozen in liquid nitrogen. Unused BCD culture medium was processed similarly as a negative control. Dry samples were weighted and stored at −80 °C until needed. For experimental analysis, dry samples were dissolved in sterile BCD medium with ammonium tartrate in the final concentration of 100 µg/µL.

### 4.4. Tests for Antibacterial Activity

Antimicrobial activity of crude moss extracts and exudates was analyzed against Gram-positive bacteria (*Staphylococcus aureus* ATCC 25923, *Streptococcus pyogenes* ATCC 12344 and *Enterococcus faecium* ATCC 35667) and Gram-negative bacteria (*Serratia marcescens SM6, Salmonella enterica ser.* Typhimurium *ATCC14028s*, *Escherichia coli TOP10* and *Pseudomonas syringae DC3000*). Strains were grown in LB medium (Gram-negative bacteria) or Tryptic soy medium (Gram-positive bacteria).

*Disk-diffusion test (DDM).* Inhibition of bacterial growth by moss metabolites was determined by the disk-diffusion method on LB agar according to CLSI guidelines (www.clsi.org, accessed on 1 July 2022). Bacterial cultures were grown overnight (ON) at 35 °C. Bacterial inoculum (CFU = 1 × 10^7^/plate) was prepared by dilution of 25 µL of ON culture in 5 mL of TOP agar (LB broth powder 25 g/L, 0.7% agar), stirred and poured on the Petri dish containing 20 mL of regular LB agar. Then, 17.5 mg of moss metabolites were added to each sterile Whatman disk (disk diameter = 7 mm). Disks soaked with 80% methanol, hexane or liquid BCD medium were used as negative controls. Disks soaked with metabolites were placed on top of inoculated plates and incubated at 35 °C for 18 h. The diameter of the bacterial growth inhibition area (halo) around each cellulose disk containing moss metabolites was then measured in mm and plotted. The diameter of the cellulose disk itself was 7 mm. All experiments were carried out in duplicate on at least three separate occasions.

*Broth microdilution method to determine minimum inhibitory concentration (MIC).* A determination of minimum inhibitory concentration (MIC) of moss metabolites was performed using 96-well microtiter plates in a BioTek plate reader spectrophotometer, the Synergy HTX [36]. Bacterial cultures were grown ON at 35 °C in MH (Mueller–Hinton) broth. Overnight cultures were subcultured at a 1:100 ratio in fresh MH broth and incubated at 35 °C with shaking (200 rpm) until each bacterial suspension reached turbidity equal to a 0.5 McFarland standard. Each resulting culture was further diluted and used to inoculate a 96-well dish containing MH broth to a final concentration of approximately 1 × 10^7^ CFU/mL. Metabolites were added using serial dilutions (50, 25, 12.5, 6.25 mg/mL or less, depending on the experiment), and the 96-well dish was sealed with a Breathe-Easy membrane (Diversified Biotech) to reduce evaporation and incubated for 18 h at 35 °C. The optical density at 600 nm (OD_600_) was measured every 15 min using a spectrophotometer (BioTek Synergy HTX). Bacterial growth in the presence of BCD medium concentrate was processed the same way as the moss exudate was used as a negative control, whereas carbenicillin and chloramphenicol dilutions were used as positive controls. Bacterial cultures grown in MH medium in the absence of metabolites were used as the general control for bacterial growth. The MIC was defined as the lowest concentration of an antimicrobial agent that inhibited the visible growth of bacteria. All experiments were performed in four biological and three technical replicates.

### 4.5. Metabolite Stability Test

Extracellular metabolites were analyzed for their stability using different treatments: boiling for 10 min, freezing/thawing, sensitivity to light and proteinase K. Lyophilized moss exudates were dissolved in MH medium and used in MIC analysis following treatments. For the freezing/thawing experiment, exudate samples were frozen in liquid nitrogen and, subsequently, thawed in 37 °C water bath three times. For the thermostability assay, exudates were incubated for 3 h at RT in a microcentrifuge tube covered with foil. For the light stability analysis, samples were exposed to white light in the transparent microcentrifuge tube for 3 h at RT. For the proteinase stability assay, samples were treated with 0.33 µg of proteinase K and incubated for 3 h at 37 °C. After treatments, all samples were subjected to MIC analysis.

### 4.6. Size Fractionation of Extracellular Moss Metabolites

The Amicon Ultra-15 ultrafiltration system (3 kDa and 10 kDa cutoff columns) (Millipore) was used to separate exudate components by their molecular weight. Fractionation was conducted following manufacturer instructions to obtain fractions <3 kDa, 3–10 kDa and >10 kDa. All procedures were performed at 4 °C.

### 4.7. Data analysis

Data were reported as mean ± standard deviation of 3 independent experiments with 2 biological replicates. Statistical significance was determined using the unpaired *t*-test with Welch’s correction; *p* < 0.05. Analyses were performed using GraphPad Prism v.9.3.1, San Diego, CA, USA.

## 5. Conclusions

This study provided new data on the antibacterial activity of extracellular and intracellular metabolites from different species of model mosses. These results highlight the applicability of mosses as a source of new bioactive compounds, as well as their biotechnological potential in medicine and agriculture. Specifically, the unique combination of *P. patens* facile genetics and genomic tools with advanced biochemical and proteomic analysis methods will make it possible to not only discover and characterize novel secondary metabolites or bioactive peptides from this previously underexplored Bryophyte model, but also to perform bioinformatics analyses to identify genes for their biosynthetic pathways.

## Figures and Tables

**Figure 1 antibiotics-11-01004-f001:**
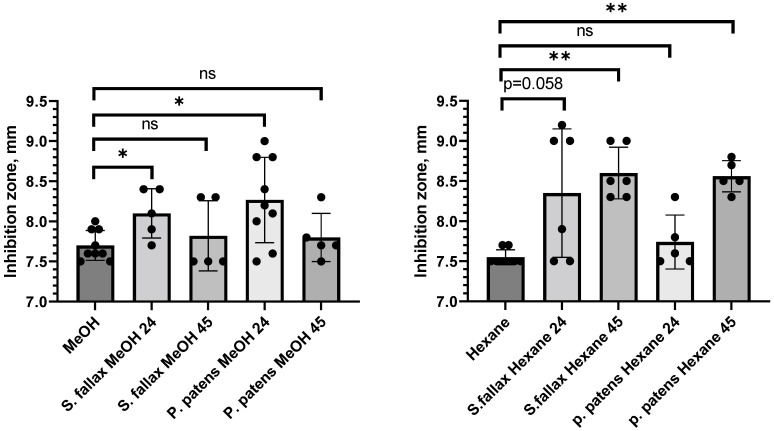
**Polar and non-polar intracellular moss metabolites inhibit the growth of *Pseudomonas syringae* DC3000.** The effect of intracellular fractions of *S. fallax* and *P. patens* mosses extracted with methanol (MeOH) (**left panel**) and hexane (**right panel**) on the growth of Gram-negative phytopathogenic *P. syringae* DC3000 was evaluated by the disk-diffusion assay (DDM); 80% methanol and hexane were used as negative controls. Moss metabolite extractions were performed for 24 or 45 h. Diameter of bacterial growth inhibition area (halo) around each cellulose disk containing moss metabolites was measured and plotted. Data represent the means from at least three independent experiments and a standard deviation. The asterisks indicate significance in an unpaired *t*-test; *—statistical significance *p* ≤ 0.05; **—statistical significance *p* ≤ 0.01; ns—not significant.

**Figure 2 antibiotics-11-01004-f002:**
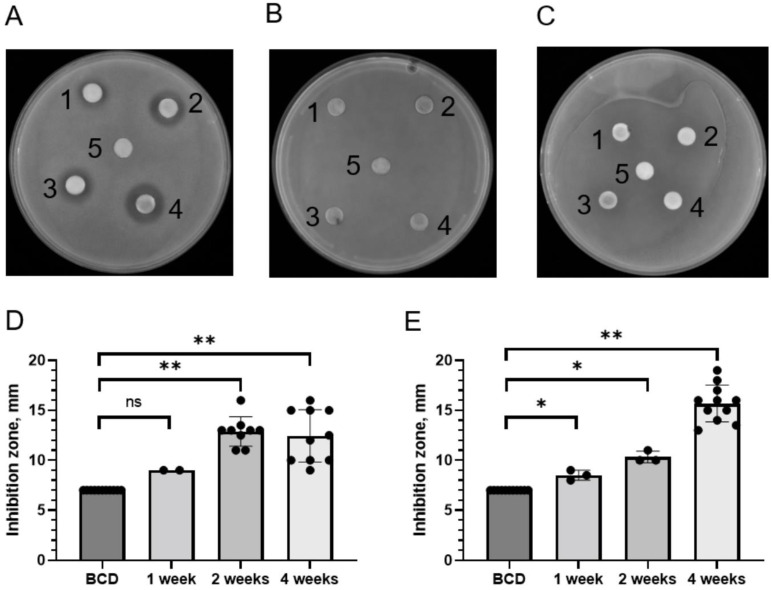
**Bacterial growth inhibitory activity of extracellular metabolites from *P. patens* ecotypes.** (**A**–**C**) Representative pictures of qualitative DDM assays with filter disks soaked with exudates from four-week-old *P. patens* Gd ecotype and placed on top of *S. aureus* ATCC25923 (**A**), *S. enterica ser.* Typhimurium ATCC14028s (**B**) and *S. marcescens* SM6 (**C**) bacterial lawns. 1–4, disks with *P. patens* exudates; 5, negative control disks (BCD medium only). (**D**,**E**) Diameter of *S. aureus* growth inhibition area (halo) around each cellulose disk containing secreted moss metabolites from one-week, two-week and four-week-old *P. patens* Gd (**D**) or Vx (**E**) ecotypes was measured and plotted. Data represent the means from at least three independent experiments and a standard deviation. The asterisks indicate significance in an unpaired *t*-test; *—statistical significance *p* ≤ 0.05; **—statistical significance *p* ≤ 0.01; ns—not significant.

**Figure 3 antibiotics-11-01004-f003:**
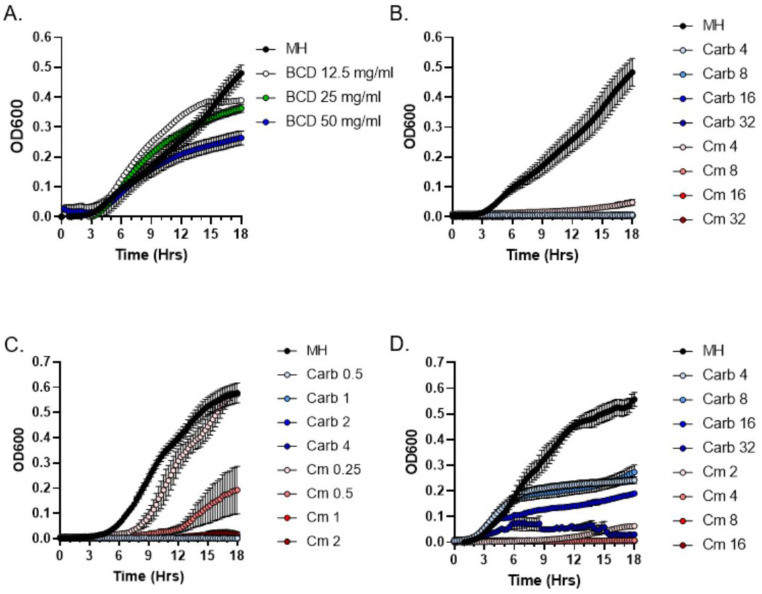
**MIC analysis of the effects of BCD medium and antibiotics carbenicillin (Carb) and chloramphenicol (Cm) on bacterial growth**. Growth curves of *S. aureus* (**A**,**B**), *Streptococcus pyogenes* (**C**) and *Enterococcus faecium* (**D**) in the presence of BCD medium (**A**) or in a range of Carb and Cm concentrations in µg/mL (**B**–**D**) are shown.

**Figure 4 antibiotics-11-01004-f004:**
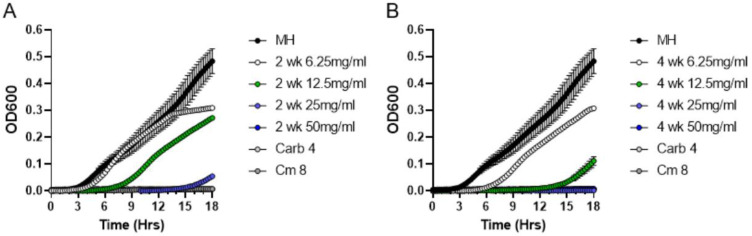
**Broth microdilution method to determine minimum inhibitory concentration (MIC) of metabolites present in *P. patens* Gd exudates**. Exudates from two-week-old (**A**) or four-week-old (**B**) *P. patens* Gd ecotype were tested in MIC assays against *S. aureus* ATCC25923. Growth curve of *S. aureus* cells was monitored in the presence of 6.25, 12.5, 25 and 50 mg/mL of exudate solution. MH—negative control, no exudate added. Carbenicillin (Carb, 4 μg/mL) and chloramphenicol (Cm, 8 μg/mL) treatments were used as positive controls for *S. aureus* growth inhibition.

**Figure 5 antibiotics-11-01004-f005:**
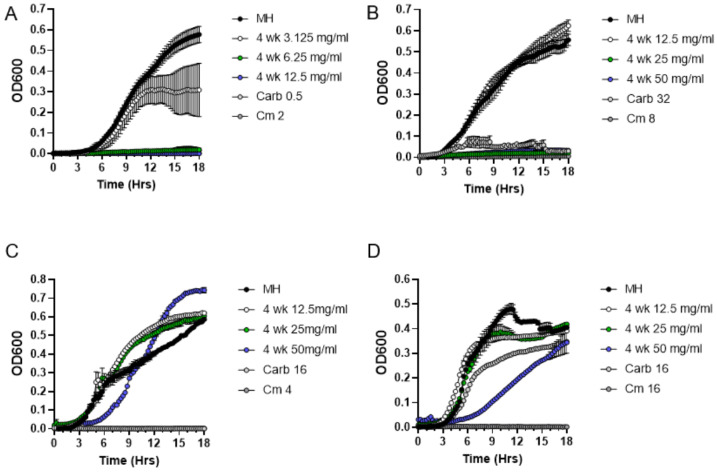
**Antibacterial activity (MIC assay) of *P. patens* exudate from four-week-old Gd strain against Gram-positive and Gram-negative bacteria**. Growth curve of *Streptococcus pyogenes* (**A**), *Enterococcus faecium* (**B**)*, Salmonella* Typhimurium (**C**) and *Serratia marcescens* (**D**) cells was monitored in the presence of different concentrations of *P. patens* exudate solution. MH—negative control with untreated bacterial cultures grown in Mueller–Hinton broth medium. Different concentrations (in μg/mL) of carbenicillin (Carb) and chloramphenicol (Cm) were used as positive controls for bacterial growth inhibition.

**Figure 6 antibiotics-11-01004-f006:**
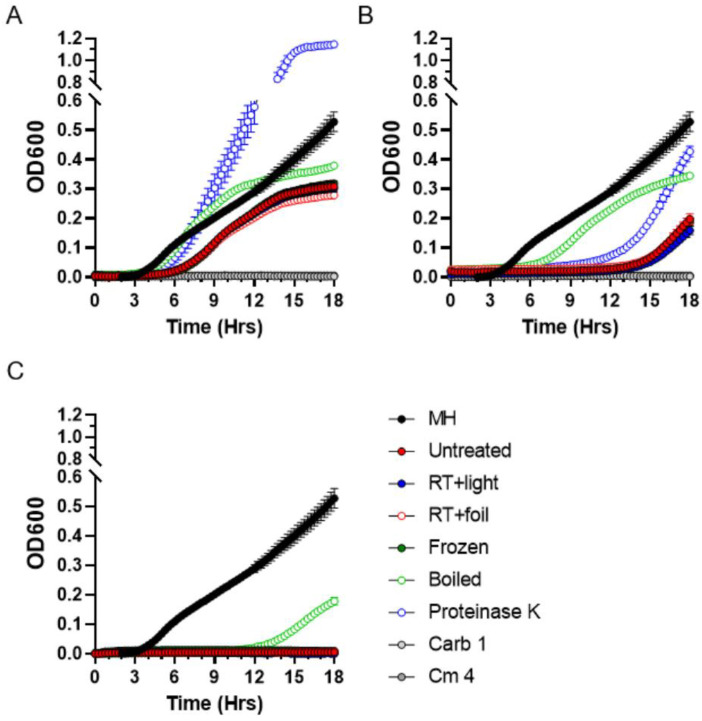
**Residual antibacterial activity of *P. patens* exudate after different treatment regimens**. Residual activity of exudate from four-week-old *P. patens* Gd culture was tested by MIC assays against *S. aureus* following treatments for temperature (Frozen, Boiled) and light/dark (RT + light, RT + foil) sensitivity, as well as proteinase K treatment. Residual activity was tested at exudate concentrations 6.25 mg/mL (**A**), 12.5 mg/mL (**B**) and 25 mg/mL (**C**). Carbenicillin (Carb, 1 µg/mL) and chloramphenicol (Cm, 4 µg/mL) were used as positive controls. MH—*S. aureus* growth in liquid MH medium without exudate addition.

**Figure 7 antibiotics-11-01004-f007:**
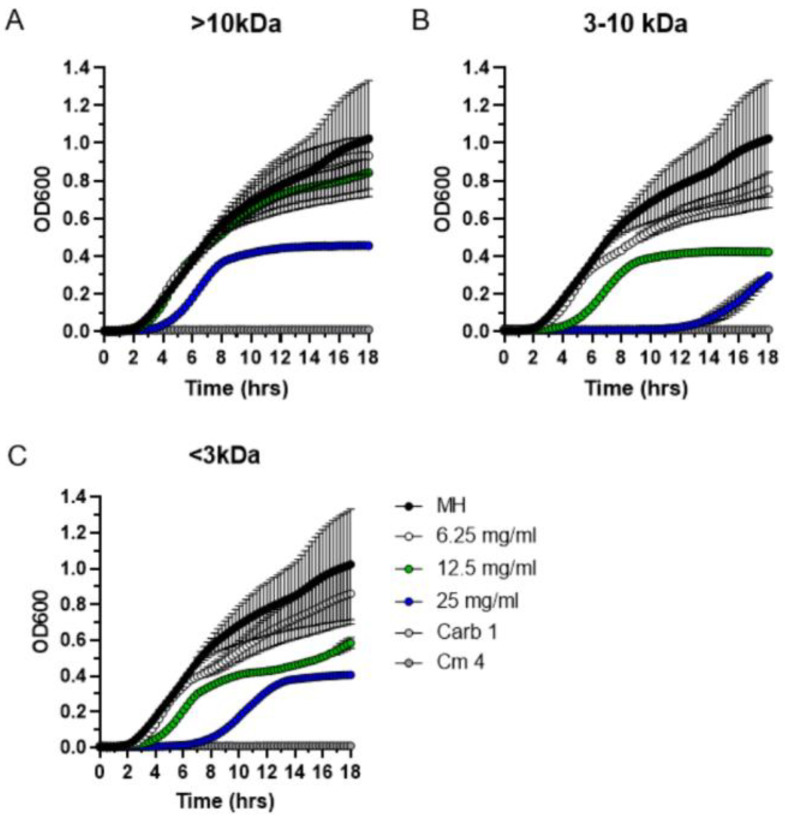
**Size fractionation of *P. patens* exudate.** Exudate from four-week-old *P. patens* Gd culture was fractionated into three molecular weight fractions, >10 kDa (**A**), 3–10 kDa (**B**), <3 kDa (**C**). Each exudate fraction was analyzed by MIC assay at three different concentrations (6.25, 12.5 and 25 mg/mL). MH—*S. aureus* growth in liquid MH medium without exudate addition. Carbenicillin (Carb, 1 µg/mL) and chloramphenicol (Cm, 4 µg/mL) were used as positive controls.

**Table 1 antibiotics-11-01004-t001:** Moss growth and intracellular metabolite extraction conditions.

Moss Species	Growth Time, Days	Extraction Solvent	Metabolite Extraction Time, h
*P. patens* Gd	10	80% methanol hexane	24, 45 24, 45
*S. fallax* MW	30	80% methanol hexane	24, 45 24, 45

**Table 2 antibiotics-11-01004-t002:** Bacterial growth inhibition activity of moss extracts in DDM test.

Moss Line	Extraction Solvent	Extraction Time, h	Growth Inhibition Zone, mm				
			Negative Control ^a^	*P. syringae* DC3000	*S. marcescens* SM6	*E. coli* TOP10	*S. aureus* ATCC25923
*P. patens* Gd	80% methanol	24	7.00 ^b^	8.30 ± 0.53 *	7.00	8.03 ± 0.59	7.00
		45	7.00	7.80 ± 0.30	7.00	8.13 ± 0.47	7.00
	Hexane	24	7.00	7.74 ± 0.34	7.00	8.37 ± 0.40	NA
		45	7.00	8.56 ± 0.19 **	7.00	8.50 ± 0.50	NA
*S. fallax* MW	80% methanol	24	7.00	8.10 ± 0.31 *	7.00	NA	NA
		45	7.00	7.82 ± 0.44	7.00	NA	NA
	Hexane	24	7.00	8.35 ± 0.80	7.00	NA	NA
		45	7.00	8.60 ± 0.32 **	7.00	NA	NA

^a^ Eighty percent methanol or hexane only; ^b^ cellulose disk diameter is 7 mm (no antibacterial activity). *—statistical significance *p* ≤ 0.05; **—statistical significance *p* ≤ 0.01; NA—not analyzed.

**Table 3 antibiotics-11-01004-t003:** Bacterial growth inhibition activity of exudates from *P. patens* ecotypes in DDM assays against *S. aureus*, in mm.

*P. patens* Ecotype	Bacterial Growth Inhibition Zone, in mm			
	No Exudate Control	1-Week-Old Moss Exudate	2-Week-Old Moss Exudate	4-Week-Old Moss Exudate
Gd	7 ^a^	9 ± 0.01	13.17 ± 1.27 **	12.97 ± 2.36 **
Vx	7	8.5 ± 0.71 *	10.33 ± 0.58 *	15.88 ± 1.65 **

^a^ Cellulose disk diameter is 7 mm (no antibacterial activity). *—statistical significance *p* ≤ 0.05; **—statistical significance *p* ≤ 0.01.

## Data Availability

The authors confirm that the data supporting the findings of this study are available within the article.

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
