# Peer review of "Antimicrobial Activities of Secondary Metabolites from Model Mosses"

_antibiotics, 2022, doi:10.3390/antibiotics11081004_

Round 1

Reviewer 1 Report

The manuscript brings new interesting knowledge about antimicrobial substances.

- references should be checked - the style of web pages citation, some old and no-so-relative articles

- p. 2 the last paragraph – from „P. patents exudates...“ is in a different font.

- at the end of the Introduction should be clearly stated the aim of the work.

- p. 14 last paragraph - different font

Author Response

Thank you very much for the review. Please see our response below.

  • References have been checked again. Two older references from 1990s were removed. The style of web pages for citations 2 and 3 has been adjusted.
  • P. patents exudates...“ have been put into correct font
  • statement on the aim of the work has been added
  • font changed for  p. 14 last paragraph

Reviewer 2 Report

The work was prepared very carefully. The introduction provides the full background and justifies the need for the research. These studies were carried out in connection with the current issues of obtaining and application of novel natural raw materials, being a source of promising secondary metabolites for the use in human health problems. Thus the study carried out by authors seems to be in line with the subjects of the journal.

Some small things – in the Materials and Methods chapter

– chapter 4.1 – the origin of the mosses used should be given

– chapter 4.2 – the mosses were ground to obtain powder – thus they had to be dried not only by paper towel, how they were dried (temperature)?

All the apparatus used in the experiments should be described (model and producer) – mortar, centrifuge, lyophilizer...

Author Response

Thank you very much for the review. Please see our response below.

  1. The origin of the moss strains is now given in chapter 4.1.
  2. Section 4.2. Tissue was blotted with paper towel to remove excess moisture, then ground up in liquid nitrogen. There was no special "drying" step involved. We now clarify this in the text to avoid confusion.
  3. Model and manufacturer information has now been added. Thank you for pointing this out.

Reviewer 3 Report

This manuscript investigated the antibacterial activity of extracellular and intracellular metabolites produced by different ecotypes (strains) of the model mosses. This is an interesting study, which highlight the applicability of mosses as a source of new bioactive compounds, and the biotechnological potential of mosses in medicine and agriculture. The experimental design is clearly descripted. In my opinion, this manuscript can be accepted after some minor revision.

1.     The words size in the manuscript is not consisted. Please set them all in the same size and style.

2.     Table 2. Please use the same format for all the numbers to display. Either same decimal places or significant digits are fine.

3.     Table 3. There are some input error for numbers in the table, such as 8,5 should be 8.5, 12,97 should be 12.97, and so on. Please check all the data carefully.

Author Response

Thank you very much for the review. Please see our response below.

  1. Manuscript font has been unified throughout the manuscript.
  2. Table 2. The same format has been implemented throughout.
  3. Table 3. Input error for numbers have been corrected. Thank you for pointing this out.